# Cu-Doped Boron Nitride Nanosheets for Solid-Phase Extraction and Determination of Rhodamine B in Foods Matrix

**DOI:** 10.3390/nano12030318

**Published:** 2022-01-19

**Authors:** Fujie Liu, Qihang Zhou, Yurui Li, Jingyu Pang

**Affiliations:** 1Zhenjiang Key Laboratory of Functional Chemistry, Institute of Medicine and Chemical Engineering, Zhenjiang College, Zhenjiang 212028, China; fjie520@163.com; 2Henan Key Laboratory of Polyoxometalate Chemistry, College of Chemistry and Chemical Engineering, Henan University, Kaifeng 475004, China; zhouqh0717@163.com (Q.Z.); L2848781123@163.com (Y.L.)

**Keywords:** adsorption, boron nitride, solid-phase extraction

## Abstract

Cu-doped boron nitride nanosheets (Cu-BNNS) were first reported as promising adsorbents for the solid-phase extraction and determination of rhodamine B (RhB) dye in a food matrix. Different characterizations, including XRD, FTIR, XPS, SEM, and TEM, were performed to confirm the formation of the adsorbent. Then, the adsorption performance of Cu-BNNS was investigated by adsorption kinetics, isotherms, and thermodynamics. Multiple extraction parameters were optimized by single-factor experiments. Under optimized conditions, the recoveries in the food matrix were in the range of 89.8–95.4%, with the spiked levels of 100 ng/mL and 500 ng/mL, respectively. This novel system was expected to have great potential to detect RhB in a wide variety of real samples.

## 1. Introduction

In the past, rhodamine B (RhB) was utilized as a coloring agent in leather, textiles, cosmetics, and other fields owing to its low cost and superior colorfastness [1,2,3]. With our increasing consciousness about food safety, RhB has been suggested to be toxic and even carcinogenic, negatively affecting food safety and directly threatening human life. Consequently, it is forbidden to add RhB into food in some regions and countries such as Europe and China [4]. However, the illegal addition of RhB to foods is still very common. As a result, high sensitivity and accuracy in detecting RhB at trace levels is essential [5].

Solid-phase extraction (SPE), a sample pretreatment method, has been widely used to pretreat pesticides, antibiotics, and other chemical products [6,7,8] due to its excellent advantages, such as high efficiency and easy operation [9]. Additionally, the adsorption efficiency of SPE mainly depends on the interactions between packing adsorbents and analytes [10,11]. Therefore, it is a significant challenge to develop adsorbents with high extraction efficiency.

Benefitting from the rapid development of synthetic chemistry and materials science, some kinds of materials, such as coordination polymer [12,13], metallic oxide [14,15], molecularly imprinted polymers [16], carbon-based materials [17], MOF [18,19], etc., have been developed as high-efficiency adsorbents. Recently, a graphene-like 2D nanomaterial, hexagonal boron nitride (*h*-BN), has been successfully applied in various processes, including hydrogen storage, gas separation, catalysis, and pollutant removal [20,21,22,23]. In addition, *h*-BN has been selected as a promising adsorbent, owing to its high mechanical strength, chemical inertness, high porosity and specific surface area, and high thermal stability [24,25,26]. Therefore, *h*-BN is an adequate candidate for solid-phase extraction [27,28].

In our previous research, element doping of *h*-BN in situ was proposed as an effective strategy for an enhanced adsorption performance. When carbon, oxygen, or silver is doped into *h*-BN, respectively, the interaction between *h*-BN and the target is increased by tuning the local electronic structure of *h*-BN [29,30,31]. With a much smaller atomic radius and lower cost than Ag, Cu is considered to be a suitable doping option. Moreover, it could also enrich the species of BN structures for the application of packing adsorbents. To the best of our knowledge, using Cu-doped BN nanosheets (Cu-BNNS) as SPE packing adsorbents has not yet been reported.

Herein, a Cu-BNNS-based SPE strategy, combined with high-performance liquid chromatography (HPLC), was investigated to concentrate and detect trace RhB in a food matrix for the first time. Firstly, Cu-BNNS were synthesized by a simple one-step method and characterized by FT-IR, XRD, XPS, SEM, and TEM, and were then employed as packing adsorbents for SPE. Then, the adsorption mechanism was discussed, including the dynamics and thermodynamics, by a static adsorption experiment. Finally, the variables that affect the extraction efficiency of SPE were optimized and applied to the determination of RhB in chili powder and drinks matrices.

## 2. Experimental Section

### 2.1. Preparation of the Adsorbents

All chemicals were of analytical-reagent grade: boron acid, urea and Cu(NO_3_)_2_, methanol, and distilled water.

### 2.2. Synthesis of Adsorbent

Boron acid, urea, and Cu(NO_3_)_2_ were added in a methanol solution with a molar ratio of 1: 30: x (x = 0.1%, 0.2%, 0.5%, 1%, 2%). The mixture was recrystallized at 55 °C. The light grey solid was obtained and calcined at 900 °C for 5 h under N_2_ atmosphere. The obtained nanomaterials were designed as Cu (x)-BN, x = 0, 0.1%, 0.2%, 0.5%, 1%, and 2%. 

### 2.3. Characterization

The FT-IR spectrum was measured from 400 to 4000 cm^−1^ on a Nexus 470 spectrometer (Nicolet Co., Ltd., Milton Freewater, OR, USA). Powder X-ray diffraction (XRD) analysis was determined using a Bruker D8 diffractometer with high-intensity Cu K α (λ = 1.54 Å) (Bruker Co., Ltd., Bremen, Germany). The chemical states of the prepared samples were carried out via X-ray photoelectron spectroscopy (XPS) by a VG MultiLab 2000 system (Thermo Fisher Scientific, Waltham, MA, USA). The morphology of samples was characterized using a JEOL JSM-7001F for field-emission scanning electron microscopy and a JEOL JEM-2010 (JEOL, Tokyo, Japan) for transmission electron microscopy.

### 2.4. Static Adsorption

Cu-BNNS (5.0 mg) were added into 25 mL RhB aqueous solutions and shaken to reach adsorption equilibrium. The concentration of RhB was detected by a UV–vis spectrophotometer (UV-2501, Shimadu, Kyoto, Japan). The adsorption capacity (*q_t_*) at different times was calculated by Equation (1): (1)$$q_{t}=\frac{{(C}_{0}-C_{t})V}{m}$$
where *C*_0_ and *C_t_* are the 0 and the *t* time concentrations of RhB solution (mg/L), *V* is the volume of RhB solution (L), and *m* is the adsorbent mass (g).

### 2.5. SPE-HPLC Procedure

A certain amount of Cu-BNNS was added into a 50 mL RhB solution (200 mg/mL) in a flask by ultrasound at room temperature. After that, the adsorbed Cu-BNNS were collected by centrifugation and eluted by methanol. Finally, the recovered solution was filtered and collected for high-performance liquid analysis. The conditions were as follows: mobile phase consisted of methanol–water (50:50, volume ratio, containing 0.3% H_3_PO_4_), the column temperature was 45 °C, the injection volume was 20 μL, the flow rate was 1 mL/min, and the characteristic peak of RhB was 545 nm.

### 2.6. Sample Matrix Preparation

First, 5 g chili powder was weighed and extracted in a flask with 100 mL ethanol at 323 K for 24 h. The supernatant was gathered and dried in N_2_ at low temperature. Then, the solid was dissolved in 50 mL methanol and stored in darkness. It was recorded as blank sample 1.

The beverage sample (Nongfu spring) was mixed with acetonitrile at the volume ratio of 1:1. The supernatant was gathered and dried in N_2_ at low temperature. Then, the solid was dissolved in 50 mL methanol and marked as blank sample 2.

## 3. Results and Discussion

### 3.1. Sorbent Characterization

#### 3.1.1. FTIR Spectrum and XRD Pattern

The FTIR spectrum of Cu(x)-BN is shown in Figure 1a. Two peaks, at 800 and 1380 cm^−1^, were ascribed to the B–N bending vibrations and B–N stretching vibrations, respectively [32,33]. The FTIR spectrum of Cu(x)-BN was very similar to that of BN. The results indicated that the BN framework remained. Figure 1b shows that two broad peaks, centered at nearly 25° and 42°, were attributed to the (002) and (100) crystal planes of BN (JCPDS no. 34-0421). Three distinct peaks, at 43°, 50°, and 74°, corresponded to the (111), (200), and (220) planes of Cu (0) (JCPDS no 01-1242) [34,35].

#### 3.1.2. XPS Spectra

XPS was used to elucidate the chemical states of Cu (5%)-BN. As shown in Figure 2a, all the elements (B, N, Cu) appear in Cu (5%)-BN. In the high-resolusion XPS spectra of Cu in Cu (5%)-BN (Figure 2b), two peaks, at 932.8 and 952.8 eV, are ascribed to Cu 2p_1/2_ and Cu 2p_3/2_, respectively [36], indicating the formation of metallic Cu. The binding energies at 190.7 and 398.4 eV were attributed to B 1s and N 1s (Figure 2c,d), respectively, indicating the existence of BN.

#### 3.1.3. SEM and TEM Image

Figure 3 illustrates the SEM (Figure 3a,b,d,e) and TEM (Figure 3c,f) images of BN and Cu(2%)-BN, which show that Cu(2%)-BN had a fluffy sheet structure similar to layered BN, and CuNP on the cover of BN can be recognized distinctly.

### 3.2. The Static Adsorption Behavior

#### 3.2.1. Adsorption Kinetics

The adsorption effect of Cu-BN with different Cu amounts for RhB were investigated (*C*_0_ = 200 ppm, *T* = 298 K). As shown in Figure 4a, the adsorption capacity rose rapidly from 0–10 min and then stabilized after 90 min. Therefore, the adsorption equilibration time was selected at 120 min. A 0.2% content of CuNP on BN (Cu (0.2%)-BN) exhibited the best adsorption capacity.

Two kinetic models, pseudo-first-order and pseudo-second-order, were used to study the adsorption behavior by Equations (2) and (3), respectively.

Pseudo-first-order kinetic model: (2)$${lg(q}_{e}-q_{t}{)\;=\;lgq}_{e}-k_{1}t\;$$

Pseudo-second-order kinetic model:(3)$$\frac{t}{q_{t}}=\frac{1}{k_{2}q_{e}^{2}}+\frac{t}{q_{e}}\;$$
where *q_e_* and *q_t_* (mg/g) are the adsorption capacity at equilibrium time and time *t*, respectively. *k*_1_ (min^−1^) and *k*_2_ (g·mg^−1^·min^−1^) are the rate constants of the pseudo-first-order model and pseudo-second-order models, respectively. The fitting curve and corresponding parameters were listed in Figure 4b and Appendix A, and the adsorption data can be fitted well with the pseudo-second-order model, according to the correlation coefficient *R*^2^ [37]. *t*_1/2_ (*t*_1/2_
*= k*_2_^−1^
*q_e_*^−1^) with a short time (2.57 min) indicated that the adsorption process had a rapid rate [38].

#### 3.2.2. Adsorption Isotherms

The adsorption isotherm, which supplies the surface properties of the solid phase, is essential to investigate the adsorbate distribution between the solid phase and the liquid phase. Langmuir and Freundlich are the two classical models to study the adsorption equilibrium.

The Langmuir isotherm model Equation (4):(4)$$\frac{C_{e}}{q_{e}}=\frac{C_{e}}{q_{m}}+\frac{1}{q_{m}K_{L}}\;$$
where *q_e_* is the equilibrium adsorption capacity (mg/g), *C_e_* is the equilibrium concentration of RhB (mg/L), *q_m_* is the theoretical maximum adsorption capacity (mg/g), and *K_L_* is a constant.

The Freundlich isotherm medel Equation (5):(5)$${lnq}_{e}{=lnK}_{F}+\frac{1}{n}\cdot {lnC}_{e}\;$$
where *C_e_* and *q_e_* are the same as defined above, *K_F_* is the Frendlich constant, and *n* is the adsorption intensity.

The Langmuir and Freundlich isotherms are shown in Figure 4c. The correlation parameters can be calculated by Equations (4) and (5), and the results are presented in Appendix A. According to the correlation coefficient *R^2^* values, the Langmuir model demonstrated a better fit than the Freundlich model, indicating a monolayer adsorption of dye on the surface of the Cu (0.2%)-BN. The calculated *q_m_* was 743 mg/g [39,40]. Several other RhB adsorbents were compared, and the results are shown in Appendix A.

#### 3.2.3. Adsorption Thermodynamics

Thermodynamics could clarify the feasibility and spontaneousness of the adsorption process, and the parameters are calculated by Equations (6) and (7).
(6)$$\Delta G=-{RTlnK}_{c}$$
(7)$${lnK}_{c}=\frac{\Delta S}{R}-\frac{\Delta H}{RT}$$

As illustrated in Appendix A, the adsorption capacity decreased with increased temperature. Low temperature facilitated the adsorption behavior within the designed temperature range.

Δ*S* and Δ*H* were calculated by Equation (7), with the linear relation of *lnK_c_* vs. 1/*T* (Figure 4d). Δ*G* was obtained by Equation (6). The detailed parameters are listed in Appendix A. The negative values of Δ*H* demonstrated the exothermic process of the adsorption reaction [41], while the negative values of Δ*G* indicated that this adsorption reaction was a spontaneous process [42,43]. The negative values of Δ*S* suggested a decrease in chaos at the liquid-solid interface in the system. Furthermore, Δ*H* also represented the reactive heat of a reaction. The smaller Δ*H* value in this system demonstrated stronger interactions between Cu-BNNS and RhB. Doping CuNP onto BN drove the B to be more electropositive, leading to the strengthening of interactions between Cu-BNNS and RhB and enhanced the adsorption capacity of BNNS [30].

### 3.3. Optimization SPE Conditions

Cu(x)-BNNS were used as a superior sorbent to extract RhB, and the extraction conditions were optimized, including adsorbent amounts, adsorption time, volume of methanol, and elute time. The results are shown in Figure 5.

#### 3.3.1. Extraction Conditions

As shown in Figure 5a, the recoveries of RhB gradually decreased with increasing amounts of adsorbents, from 5 to 30 mg. It can be found that excessive extractant did not facilitate the recovery of RhB. Considering the cost, the optimal mass of a solid-phase adsorbent was 5 mg. As indicated in Figure 5b, the recoveries of RhB increased to 86.5% with the increase in time (6.0–8.0 min), and it fell to around 69.4% when the adsorption time was increased to 16 min. Thus, 8.0 min was selected as the most favorable adsorption time.

#### 3.3.2. Elution Conditions

As illustrated in Figure 5c, when the volume of methanol reached 5 mL, the recovery of RhB was 87.2%, indicating that most of the RhB under this volume was elution. When the volume of methanol was increased to 5.5 mL, the recovery rate maintained stability. Therefore, 5 mL was the volume of methanol selected in this experiment. To achieve rapid desorption, the saturated Cu(0.2%)-BN was also placed in methanol under ultrasonic conditions, and the results are shown in Figure 5d. As the elution time increased, the recovery of RhB increased gradually. When the elution time reached 11 min, the best effect was achieved, and the recovery rate reached 92.0%. With the increase in ultrasonic elution time, the recovery remained stable, and most of the RhB was eluted completely. A total of 11 min of elution time was selected as the optimum condition.

### 3.4. Analysis of Spiked Samples

An optimized condition was used to recover RhB in chili powder and beverage sample matrix. No RhB was detected in blank samples 1 and 2. Appendix A illustrates the chromatogram of blank sample 1 and the spiked sample (500 ng/mL). The peak of the RhB was detected at 9.5 min after the addition of the standard sample. The analytical results are shown in Table 1. The recoveries of RhB in food samples by this method were 89.8–95.4%, and the RSD value was less than 4.3. Several other RhB detection methods were compared by referring to the literature. The results are shown in Appendix A and indicate that this method is suitable for the detection of RhB in chili powder and beverages.

## 4. Conclusions

In this study, Cu-BNNS were prepared as an efficient adsorbent for the extraction and determination of RhB, using SPE in food samples. Static adsorption behavior indicated that RhB could be rapidly adsorbed on the Cu-BNNS with *t*_1/2_ = 2.57 min, the maximum adsorption capacity of RhB was 743 mg/g by the Langmuir isotherm, and this adsorption system is an exothermic and spontaneous process. Furthermore, the optimum SPE conditions were: extraction time, 8 min; amount of C-BNNS, 25 mg; elution volume of methanol, 5 mL; and elution time, 11 min. A perfect spiked recovery of 89.8–95.4% was also obtained under the optimum conditions for analyzing RhB in a food matrix.

## Figures and Tables

**Figure 1 nanomaterials-12-00318-f001:**
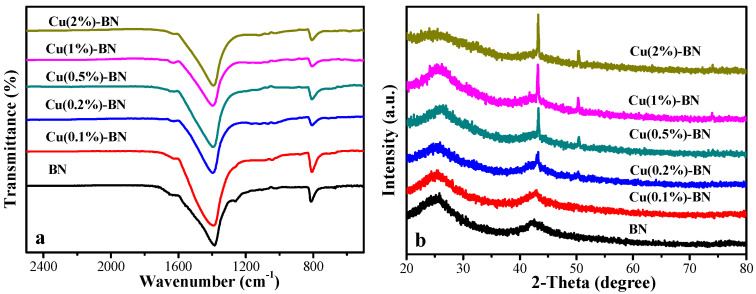
(**a**) FTIR spectra; (**b**) XRD patterns of Cu(x)-BN, x = 0, 0.1%, 0.2%, 0.5%, 1%, and 2%.

**Figure 2 nanomaterials-12-00318-f002:**
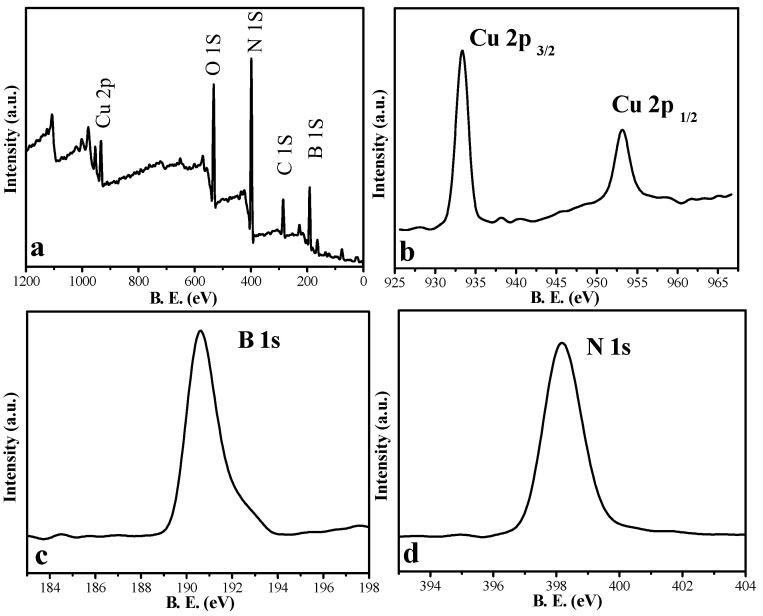
(**a**) XPS survey spectra of Cu (5%)-BN, high-resolution XPS spectra of (**b**) Cu 2p, (**c**) B 1s, and (**d**) N 1s of samples.

**Figure 3 nanomaterials-12-00318-f003:**
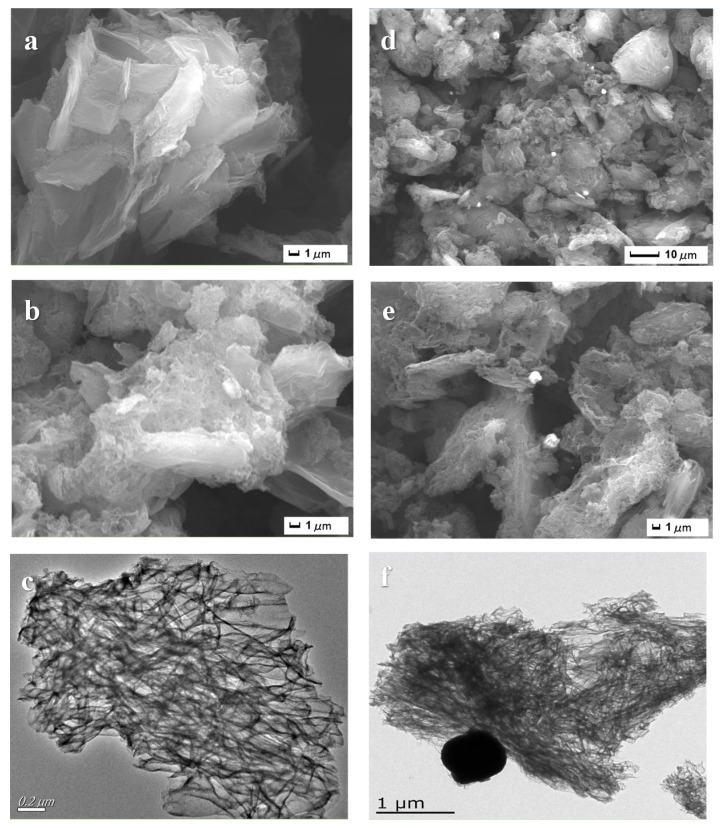
SEM images of BN (**a**,**b**), SEM images of Cu(2%)-BN (**d**,**e**), TEM images of BN (**c**), Cu(2%)-BN (**f**).

**Figure 4 nanomaterials-12-00318-f004:**
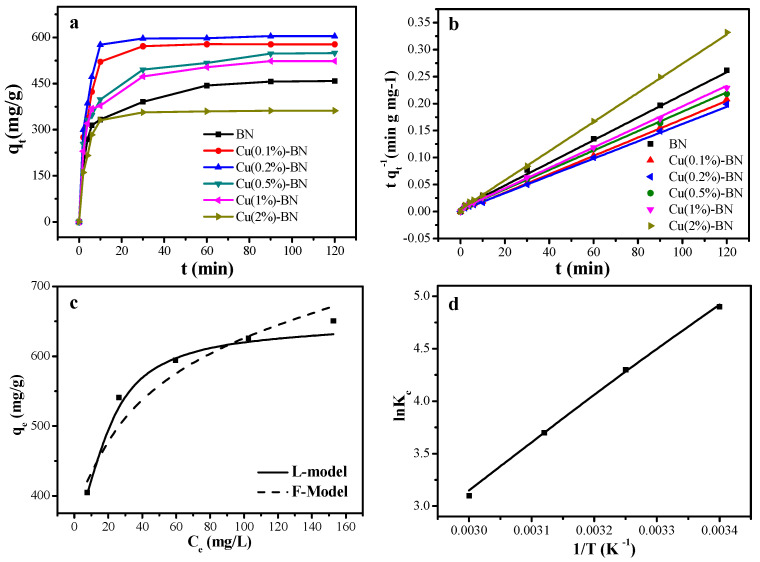
(**a**) Effect of time on the *q_t_* of RhB on Cu(x)-BNNS, where x = 0, 0.1%, 0.2%, 0.5%, 1%, and 2%. (**b**) The plots of the kinetic model (pseudo-second-order) of RhB onto Cu(x)-BN. (**c**) Langmuir (L-model) and Freundlich (F-model) adsorption isotherm models of RhB on Cu(0.2%)-BNNS. (**d**) Van’t Hoff regressions of RhB on Cu(0.2%)-BNNS.

**Figure 5 nanomaterials-12-00318-f005:**
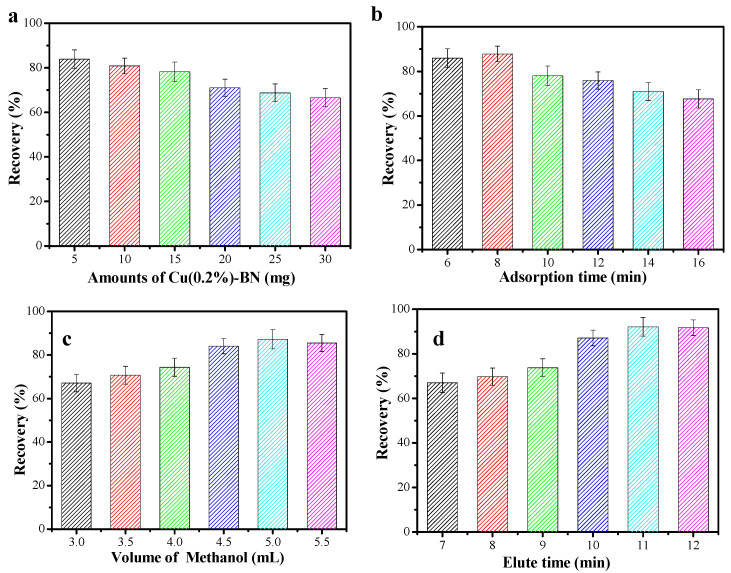
Effect of (**a**) different amounts of Cu(0.2%)-BN, (**b**) adsorption time, (**c**) elution volume, and (**d**) elution time on the recovery of RhB during the SPE procedure.

**Table 1 nanomaterials-12-00318-t001:** Determination results of the samples (*n* = 3).

Sample	Added (ng/mL)	Recovery (%)	RSD (%, *n* = 3)
Chili powder	100	91.0	4.3
	500	89.8	3.1
Drinks	100	92.1	3.0
	500	95.4	3.8

## Data Availability

The data presented in this study are available on request from the corresponding author.

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
