# Peer review of "Cu-Doped Boron Nitride Nanosheets for Solid-Phase Extraction and Determination of Rhodamine B in Foods Matrix"

_nanomaterials, 2022, doi:10.3390/nano12030318_

Round 1

Reviewer 1 Report

Wide utilization of chemicals in food production and packaging industry requires that these additives are safe to the human health. However, very often they are not. Even very small amounts of them may be harmful, moreover when they accumulate. This requires the ways, often at the state-of-the art level, to detect and remove such undesirable additives. In relation to this, the present work reporting the effect of synthesized and thoroughly characterized Cu doped boron nitride nanosheets for solid-phase extraction and determination of Rhodamine B in foods matrix, is certainly in need. Adsorption characteristics of synthesized Cu-BN NS are determined, demonstrating its quite high sensitivity and effectiveness as an adsorbent for RhB. The strength of the work: Precise sample preparation routes in combination with appropriate complimentary characterization tools enabling quite thorough and unambiguous results. Quite immediate applied relevance of the work.  The weakness: To more precisely asses the effect of Cu additive and its interaction with BN upon the adsorption characteristics, it would be useful to determine the charge (neutral or partly ionic) state of Cu, for instance by direct comparison of XPS 2p line position and shape of Cu-BN against the metallic Cu (located in the chamber for reference).  In general, the work is scientifically sound, logically arranged and clearly presented. Figures are informative and clear; reference list is comprehensive and up-to-dated. In my view, the manuscript is suitable for publication in Nanomaterials in its present form. My only optional recommendation is minor language editing throughout the text, for instance, lines 111-112 “which indicated that the metallic form of Cu.” (I guess needs to be continued); lines 193-194 (fallen to have fallen, fell), etc.

Author Response

Answer: Thank the reviewer for the sincere comments. In my opinion, the interaction between Cu and BN maybe adsorption force. There are some defect sites during the formation of BN, then Cu nanoparticle was adhered by solid adsorption.

For the valence state of Cu, we think the major component maybe metallic Cu. UV–Vis DRS spectra were also employed. It is noted that a new peak around 570 nm can be detected in Cu(2%)-BN which is assigned to Surface Plasmon Resonance (SPR) peak of copper, also proving the hybrid of Cu NPs and BN, which is in consistent with XRD and XPS analysis.

Language editing was also revised. lines 111-112 was changed and relative descriptions were revised as follows: “indicating the formation of metallic Cu”; lines 193-194 “fallen” changed to “fell”.

The revised parts were marked with a blue background.

Fig. R1. UV–Vis DRS spectra of g-BN and Cu NPs/g-BN.

Reviewer 2 Report

Liu and his colleagues in their study report the synthesis of Cu-BNNS for adsorption and extraction of RhB using SPE in food samples. It is interesting to see that RhB could be rapidly adsorbed on Cu-BNNS in short time within t1/2=2.57 min with a maximum adsorption capacity of RhB was 743 mg/g. The manuscript is well-written, organized, and English is acceptable and can be considered for publications in Nanomaterials after figuring out the following comments throughout their manuscript:

  1. I wonder why the author used this big amount of urea.
  2. It would be better if the author assigns FTIR spectra and XRD patterns in Fig.1a and b.
  3. Why did the authors use Cu(5%)-BN for XPS and on the other hand use Cu(2%)-BN for SEM and TEM characterization?
  4. I think the caption in Fig. 3 is wrong, please revise it.
  5. Where is Fig. S1 and Tables S1/S2.
  6. Where is the interpretation inside the text of adsorption isotherms?
  7. Why did the Cu(2%)-BN show the best adsorption efficiency?
  8. Where is Fig. S3? I am not able to find the supporting information file.

Author Response

  1. I wonder why the author used this big amount of urea.

Answer: When the temperature was increased unceasingly, urea with a low crystal surface energy may pyrolyse sequentially and produces numerous gases such as NH3, CO2 and H2O to fabricate micro/mesopores in the already-formed BNNs. So, the role of big amount of urea was pore agents.

  1. It would be better if the author assigns FTIR spectra and XRD patterns in Fig.1a and b.

Answer: Thank you for your suggestion. We have changed as follows:

“Fig.1. (a) FTIR spectra; (b) XRD patterns of Cu(x)-BN, x = 0, 0.1%, 0.2%, 0.5%, 1%, 2%

Fig.1. (a) FTIR spectra; (b) XRD patterns of Cu(x)-BN, x = 0, 0.1%, 0.2%, 0.5%, 1%, 2%.”

  1. Why did the authors use Cu(5%)-BN for XPS and on the other hand use Cu(2%)-BN for SEM and TEM characterization?

Answer: As we all know, XPS signals can only come from a very few atom layers from the surface. The signal from Cu may not be distinctly observed if its content is low. The Cu signal of Cu(2%)-BN was not detect. According to the XRD patterns, with the increase of Cu, the degree of crystallization was enhanced only. So Cu(5%)-BN was chosen to analyse.

  1. I think the caption in Fig. 3 is wrong, please revise it.

Answer: Thanks. We check the number and revised the relative part.

Fig. 3. SEM images of BN (a, b) and Cu(2%)-BN (d, e), TEM images of BN (c) and Cu(2%)-BN (f).

  1. Where is Fig. S1 and Tables S1/S2.

Answer: Thank. We check the relative part. Tables S1/S2 were in line 148 and 165. Fig. S1 was in line 174. And the “Supporting Information” was also checked carefully.

  1. Where is the interpretation inside the text of adsorption isotherms?

Answer: Thanks. The interpretation was in Line 166-167. “Langmuir model demonstrated better fit than the Freundlich model, indicating a monolayer adsorption of dye on the surface of Cu (0.2%)-BN”. The adsorption isotherm, supplying the surface properties of solid, is essential to investigate the adsorbates distribution between solid phase and liquid phase. Langmuir model clarified a monolayer adsorption while Freundlich model was a polymolecular layer adsorption. According to the correlation coefficient R2 values of different model, we can judge which of them fitting better 1,2.

  1. Why did the Cu(2%)-BN show the best adsorption efficiency?

Answer: Thanks. In line 138, “0.2% content of CuNP on BN (Cu (0.2%)-BN) exhibited the best adsorption capacity.” In Fig. 4a, it can be found that suitable decorated content of 0.2% Cu on BN exhibited the best adsorption capacity. In our opinion, the enhancement of adsorption capacities was due to the interfacial coupling between Cu and BN with the introduction of appropriate content Cu onto BN 3. The decrease of adsorption capacity for more than 0.2% of Cu may be caused by the aggregation of Cu nanoparticles and the sharply declined specific surface area.

  1. Where is Fig. S3? I am not able to find the supporting information file.

Answer: Thanks. Sorry for our carelessness. The Fig. S3 should be numbered Fig. S2.

Reference

  1. Prasannamedha, G.; Kumar, P.S.; et al. Enhanced adsorptive removal of sulfamethoxazole from water using biochar derived from hydrothermal carbonization of sugarcane bagasse. J. Hazard. Mater., 2021, 407, 124825.
  2. Wang, H.F.; Li, Z.C.; et al. Effective adsorption of dyes on an activated carbon prepared from carboxymethyl cellulose: Experiments, characterization and advanced modelling. Chem. Eng. J., 2021, 417, 128116.
  3. Pang, J.; Chao, Y.; et al. Silver Nanoparticle-Decorated Boron Nitride with Tunable Electronic Properties for Enhancement of Adsorption Performance. ACS Sustain. Chem. Eng. 2018, 6, 4948-4957.

Reviewer 3 Report

The manuscript is deal with investigation Cu-BNNS based SPE strategy combined with High Performance Liquid Chromatography (HPLC) to concentrate and detect trace RhB in food matrix. This work represents a novel idea and good presented, however, the manuscript required minor modifications to be improved.

1- English Language should be revised all over the manuscript to avoid grammatical mistakes.

2- The author stated at materials and method section (line 71) that EDX analysis was done for the prepared material, however, result and discussion section not include this result, it should be adjusted at the manuscript and justified.

3- What is the expected interaction mechanism between Cu-BNNS and RhB dye. it should be mentioned.

4- the adsorption results of Cu-BNNS and RhB dye should be compared with literature.

4- The reference list should be updated, where, some new related references were missed such as 

-doi.org/10.3390/ma14247545.

- doi.org/10.3390/polym13132033

- doi.org/10.3390/polym 13223869.

Author Response

  • English Language should be revised all over the manuscript to avoid grammatical mistakes.

Answer: We have scrutinized our manuscript very carefully and minimize the errors of typo and grammar in the revised paper. The revised parts were marked with a blue background.

  • The author stated at materials and method section (line 71) that EDX analysis was done for the prepared material, however, result and discussion section not include this result, it should be adjusted at the manuscript and justified.

Answer: Thanks for the valuable suggestion. The “2.3. Characterization” was checked carefully and the relative changed were revised in manuscript.

“FT-IR spectrum was measured from 400 to 4000 cm-1 on a Nexus 470 spectrometer. Powder X-ray diffraction (XRD) analysis was determined using a Bruker D8 diffractometer with high-intensity Cu K α (λ =1.54 Å). The chemical states of the prepared samples were carried out via X-ray photoelectron spectroscopy (XPS) by a VG MultiLab 2000 system. The Cu-BN morphology have been characterized by using a JEOL JSM-7001F for field-emission scanning electron microscope and JEOL JEM-2010 for Transmission electron microscopy”

  • What is the expected interaction mechanism between Cu-BNNS and RhB dye. it should be mentioned.

Answer: Thanks. The interaction mechanism has been discussed in Line 182-187. “The smaller ∆H value in this system demonstrated stronger interactions between Cu-BNNS and RhB. Doping CuNP onto BN drived the B more electropositive, leading to the strengthen of interactions between Cu-BNNS and RhB and enhanced the adsorption capacity of BNNS”

  • the adsorption results of Cu-BNNS and RhB dye should be compared with literature.

Answer: Thanks for the valuable suggestion. The literature had been listed in Supporting Information.

Table S3 Comparison of the BN adsorption capacity of RhB with other literatures.

Adsorbents

qm

Ref.

cotton flower-like hierarchically porous boron nitride

313.4

[1]

hexagonal boron nitride nanosheets

16

[2]

Porous hexagonal boron nitride whiskers

210.1

[3]

boron nitride nanosheets

75.65

[4]

Few-Layer Boron Nitride

488

[5]

Flake Boron Nitride

125

[6]

boron nitride nanosheets

124

[7]

Cu doped boron nitride nanosheet

743

This work

qm: the theoretical maximum adsorption capacity calculated by adsorption isotherms model.

4- The reference list should be updated, where, some new related references were missed such as 

-doi.org/10.3390/ma14247545.

- doi.org/10.3390/polym13132033

- doi.org/10.3390/polym 13223869.

Answer: Thanks. We added the three references in Introduction part.

Reference

  1. Maiti, K., Thanh, T. D., Sharma, K., Hui, D., Kim, N. H., and Lee, J. H., Highly efficient adsorbent based on novel cotton flower-like porous boron nitride for organic pollutant removal. Composites Part B-Engineering 2017. 123, 45-54.
  2. Mahdizadeh, A., Farhadi, S., and Zabardasti, A., Microwave-assisted rapid synthesis of graphene-analogue hexagonal boron nitride (h-BN) nanosheets and their application for the ultrafast and selective adsorption of cationic dyes from aqueous solutions. Rsc Advances 2017. 7, 53984-53995.
  3. Li, Q., Yang, T., Yang, Q. F., Wang, F., Chou, K. C., and Hou, X. M., Porous hexagonal boron nitride whiskers fabricated at low temperature for effective removal of organic pollutants from water. Ceramics International 2016. 42, 8754-8762.
  4. Wang, X. B., Yang, Y. F., Jiang, G. D., Yuan, Z. W., and Yuan, S. D., A facile synthesis of boron nitride nanosheets and their potential application in dye adsorption. Diamond and Related Materials 2018. 81, 89-95.
  5. Chang, H. H., Chao, Y. H., Pang, J. Y., Li, H. P., Lu, L. J., He, M. Q., Chen, G. Y., Zhu, W. S., and Li, H. M., Advanced Overlap Adsorption Model of Few-Layer Boron Nitride for Aromatic Organic Pollutants. Industrial & Engineering Chemistry Research 2018. 57, 4045-4051.
  6. Qu, J. L., Li, Q., Lu, C., Cheng, J., and Hou, X. M., Characterization of Flake Boron Nitride Prepared from the Low Temperature Combustion Synthesized Precursor and Its Application for Dye Adsorption. Coatings 2018. 8.
  7. Chao, Y., Liu, M., Pang, J., Wu, P., Jin, Y., Li, X., Luo, J., Xiong, J., Li, H., and Zhu, W., Gas-assisted exfoliation of boron nitride nanosheets enhancing adsorption performance. Ceramics International 2019. 45, 18838-18843.
